# γδ T Cells in Glioblastoma Multiforme: Novel Roles and Therapeutic Opportunities

**DOI:** 10.3390/cancers17162660

**Published:** 2025-08-15

**Authors:** Costanza Dieli, Rosario Maugeri, Anna Maria Corsale, Marta Di Simone, Claudia Avellone, Francesco Dieli, Domenico Gerardo Iacopino, Lara Brunasso, Alessandra Cannarozzo, Roberta Costanzo, Silvana Tumbiolo, Serena Meraviglia

**Affiliations:** 1Central Laboratory of Advanced Diagnosis and Biomedical Research (CLADIBIOR), University of Palermo, 90133 Palermo, Italy; costanza.dieli@unipa.it (C.D.); annamaria.corsale@unipa.it (A.M.C.); marta.disimone@unipa.it (M.D.S.); claudia.avellone@unipa.it (C.A.); alessandra.cannarozzo@community.unipa.it (A.C.); serena.meraviglia@unipa.it (S.M.); 2Department of Precision Medicine in Medical, Surgical and Critical Care (MePreCC), University of Palermo, 90133 Palermo, Italy; 3Department of Biomedicine, Neuroscience and Advanced Diagnosis (BiND), University of Palermo, 90133 Palermo, Italy; rosario.maugeri@unipa.it (R.M.); gerardo.iacopino@unipa.it (D.G.I.); l.brunasso@villasofia.it (L.B.); r.costanzo@villasofia.it (R.C.); 4Department of Health Promotion, Mother and Child Care, Internal Medicine and Medical Specialties (ProMISE), University of Palermo, 90133 Palermo, Italy; 5Neurosurgery Unit, “Villa Sofia” Hospital, 90146 Palermo, Italy; silvana.tumbiolo@villasofia.it

**Keywords:** γδ T lymphocytes, glioblastoma, immunotherapy

## Abstract

Glioblastoma multiforme is one of the most aggressive brain tumors, with limited treatment success due to its ability to suppress the immune system and resist conventional therapies. This review explores the emerging role of a unique subset of immune cells, known as γδ T cells, which can identify and destroy tumor cells without relying on traditional antigen presentation pathways. The authors aim to examine how these cells function within the glioblastoma environment and evaluate their potential as a novel immunotherapeutic approach. By highlighting both the opportunities and challenges associated with γδ T cell-based strategies, this research may support the development of innovative therapies that improve outcomes for patients with glioblastoma and potentially other solid tumors.

## 1. Introduction

### 1.1. Glioblastoma and the Immunosuppressive Microenvironment

GBM is the most common and lethal primary brain tumor, accounting for nearly 45% of all primary malignant brain tumors, with a median survival of less than one year, despite aggressive treatments such as surgical resection, radiotherapy, and chemotherapy (also known as Stupp’s treatment). It is characterized by an incidence of approximately 3 per 100,000 people annually [1,2]. Despite advances in surgical techniques and the use of chemotherapeutic agents like temozolomide (TMZ), the recurrence of GBM remains inevitable due to its heterogeneity and the lack of effective therapeutic strategies [3]. A key factor contributing to the poor prognosis of GBM is its immunosuppressive tumor microenvironment (TME), which prevents the immune system from mounting an effective antitumor response. Tumor-associated macrophages (TAMs), myeloid-derived suppressor cells (MDSCs), and regulatory T cells (Tregs) dominate the TME, creating a hostile environment that supports tumor growth while suppressing immune function [4]. The tumor’s highly infiltrative nature and resistance to conventional therapies contribute significantly to the challenges in achieving long-term survival for patients. Additionally, the blood–brain barrier (BBB) restricts the delivery of many therapeutic agents, further complicating treatment options [5]. This makes developing novel and effective immunotherapies a critical area of research in the fight against GBM.

### 1.2. Immune Checkpoint Inhibitors in GBM

One promising avenue of immunotherapy involves immune checkpoint inhibitors (ICIs), which have revolutionized the treatment of several cancers, but their impact on GBM has been limited [6]. Several ICIs have been tested for their efficacy in GBM treatment, among which are those targeting PD-1 (Programmed Death-1) and PD-L1 (Programmed Death Ligand-1), as well as CTLA-4 (Cytotoxic T-Lymphocyte Antigen-4). Pembrolizumab (Keytruda), which targets PD-1, has been explored as a potential treatment for GBM, particularly when combined with other therapeutic approaches. Similarly, Nivolumab (Opdivo), another PD-1 inhibitor, has also been investigated in clinical trials for its efficacy on GBM, although the results have been mixed [7]. Atezolizumab (Tecentriq), which inhibits PD-L1, has been studied in various cancers, including brain tumors, and its effects on GBM have been closely examined [8]. Additionally, Ipilimumab (Yervoy), a CTLA-4 inhibitor primarily used in melanoma, has been tested for GBM treatment, often in combination with other ICIs such as nivolumab [9]. Despite the initial promise surrounding these treatments, the overall effectiveness of ICIs in treating GBM remains limited: although ICIs such as pembrolizumab, nivolumab, and ipilimumab have shown positive results in other types of cancer, their impact on GBM is less pronounced. This is mainly due to GBM’s low mutational burden, which results in insufficient immune cell activation within the TME [10]. Furthermore, the presence of an immunosuppressive microenvironment, coupled with the low levels of immune infiltration typically found in GBM, reduces the efficacy of ICI therapies [11].

### 1.3. γδ T Cells as a Novel Immunotherapeutic Strategy

An emerging approach to GBM immunotherapy involves the use of γδ T cells, a subset of T lymphocytes with innate-like properties that make them particularly effective in recognizing and eliminating tumor cells. γδ T cells do not require the classical antigen presentation through major histocompatibility complex (MHC) molecules, which allows them to target a broader range of tumor antigens, including those expressed on GBM cells [12]. Interestingly, recent studies have shown that γδ T cells do infiltrate gliomas, including GBM. One study reported that γδ T cells accounted for an average of 1.7% of tumor-infiltrating immune cells in GBM [13]. Another study quantified γδ T cells at 1.29% in GBM and 1.38% in low-grade glioma, compared to only 0.8% and 0.49%, respectively, for conventional αβ T cells [14]. These findings highlight the relative abundance and potential relevance of γδ T cells within the glioma microenvironment. Recent studies suggest that γδ T cells can be activated in vivo or expanded ex vivo to enhance their antitumor activity against GBM [15]. These cells can directly kill tumor cells, modulate the immune response, and produce pro-inflammatory cytokines that can help overcome the immunosuppressive TME of GBM [16]. Furthermore, the ability of γδ T cells to interact with other immune cells, such as dendritic cells and natural killer (NK) cells, provides a unique advantage in promoting a more robust and coordinated immune response against the tumor [16]. γδ T cells can exhibit pro- and anti-tumoral effects depending on their activation state and the surrounding microenvironment, and these cells are crucial in counteracting the immunosuppressive milieu created by TAMs, MDSCs, and Tregs [12]. As a result, researchers are increasingly exploring γδ T cells as a potential therapeutic strategy, either as an adjunct to traditional treatments or in combination with other immunotherapies [16]. This review highlights the role and clinical efficacy of using γδ T cells for GBM treatment. We will explore their role in the immune response to GBM, the mechanisms underlying their antitumor activity, and the clinical studies investigating their potential in overcoming the challenges posed by the GBM microenvironment.

## 2. γδ T Cell Characteristics and Functions

γδ T cells represent a unique subset of CD3^+^ T lymphocytes, classified as “unconventional” T cells due to their distinct T cell receptor (TCR) structure. Unlike conventional αβ T cells, γδ T cells express a TCR composed of γ and δ chains and account for approximately 1–5% of the total peripheral T lymphocytes in the human blood [17]. γδ T cells bridge innate and adaptive immunity, providing rapid immune responses in collaboration with macrophages and neutrophils against various pathogens. Additionally, they assist in modulating the function of adaptive immune cells, including T cells and B cells [18]. Notably, γδ T cells express receptors typical of NK cells, such as NKG2D, CD94-NKG2A/C, and the natural cytotoxicity receptors (NCRs) NKp30 and NKp44 [19], which enable them to target and kill infected, stressed, or neoplastic cells and to activate other immune effectors [18]. Most γδ T cells (around 70%) lack both CD4 and CD8 expression (CD4^−^CD8^−^), while about 30% are CD8^+^CD4^−^, and fewer than 1% are CD4^+^CD8^−^. They are categorized into three main subsets based on their γ and δ chain composition: Vδ1^+^, Vδ2^+^, and Vδ1^−^Vδ2^−^. The Vδ1^+^ T cells are predominantly localized in mucosal and epithelial tissues, where they exhibit cytotoxic activity through the recognition of stress-induced antigens expressed by transformed cells, such as MICA/MICB and UL16-binding proteins, via NKG2D [16] or NCRs [19]. The Vδ1 chain can associate with various γ chains, including Vγ2, Vγ3, Vγ4, Vγ5, Vγ8, and Vγ10, with Vγ4 showing a notable preferential pairing with Vδ1 in intestinal tissue. The most abundant subset of γδ T cells expresses the Vδ2 chain, commonly paired with Vγ9, forming what are known as Vγ9Vδ2 T cells. These cells account for 50–95% of the total γδ T lymphocyte population and are characterized by their ability to respond to phosphoantigens (PAgs). PAgs are intermediates of the mevalonate metabolic pathway, which is upregulated in tumor cells, or the exogenous 1-deoxy-D-xylulose-5-phosphate (MEP) pathway, which is enhanced during infections [20]. The mevalonate pathway, essential for the biosynthesis of cholesterol, sterols, steroid hormones, and cellular membranes, generates PAgs as intermediates, which serve as signals for γδ T cells to distinguish between normal and abnormal (tumor or infected) cells [21]. High levels of PAgs indicate cellular stress or transformation, and promote γδ T cells activation to target and eliminate these altered cells [22]. PAgs-induced γδ T cell activation involves two butyrophilins, BTN2A1 and BTN3A1. The BTN2A1–BTN3A1 complex interacts with the γδ TCR at two distinct sites: BTN2A1 binds to the Vγ9 domain, while BTN3A1 binds to the Vδ2 regions and the γ chains of the TCR [23]. Once activated, γδ T cells target infected, stressed, or tumor cells using mechanisms such as cytotoxicity via perforin and granzyme secretion, antibody-dependent cellular cytotoxicity (ADCC), and the release of pro-inflammatory cytokines like TNF-α and IFN-γ. These mechanisms are also employed by NK and CD8^+^ T cells to eliminate abnormal cells [24]. Human γδ T cells, particularly the Vγ9Vδ2 subset, exhibit significant phenotypic and functional heterogeneity, allowing them to adopt various differentiation states that influence their responsiveness and antitumor potential and that, in turn, can be influenced by the TME (Figure 1) [25]. Understanding this diversity is crucial for the development of γδ T cell-based immunotherapies, as the differentiation state significantly impacts their cytotoxic potential, tissue distribution, and persistence within the TME [26].

## 3. Human γδ T Cells in GBM

γδ T cells are involved in the immune response against tumors [27]. These cells are well known for their ability to recognize and target tumor cells in various malignancies, including GBM [13,16]. In GBM, γδ T cells are recruited into the TME, where they can interact with tumor cells and other immune cells, including macrophages and dendritic cells [25]. These cells have been found to infiltrate GBM tissues in mouse models [16,28] and human patients [16,25], with their infiltration correlating with enhanced survival and tumor regression. Moreover, studies have demonstrated that γδ T cells, particularly those expressing the Vγ9Vδ2 TCR, play a key role in immune surveillance in GBM, exhibiting both cytotoxic activity and the capacity to produce pro-inflammatory cytokines that can enhance the immune response to tumor cells [28]. It was observed that Vγ9Vδ2 γδ T cells exhibit potent cytotoxic activity against human glioma cell lines and GBM tumor cells, suggesting their potential for therapeutic applications in GBM [27,29]. However, despite these promising findings, the role of γδ T cells in GBM remains complex. Some studies suggest that these cells may enhance antitumor immunity, while others have indicated that the immunosuppressive TME in GBM can impair their effectiveness [30]. This is partly due to the functional plasticity exhibited by γδ T cells, allowing them to differentiate into different subsets based on the microenvironmental signals present within the tumor such as the cytokine milieu and interactions with other immune and non-immune cells [4,31].

## 4. Anti-Tumoral Role of γδ T Cells in GBM

γδ T cells have been shown to play a critical anti-tumoral role in GBM. An analysis of the Cancer Genome Atlas (TCGA) database has shown that γδ T cell activity positively correlates with CD8^+^ cytotoxic T cell activity and M1 macrophage presence, highlighting their potential to stimulate the immune response against GBM [32]. One of the most well-characterized mechanisms by which γδ T cells mediate anti-tumor responses is their ability to recognize and kill tumor cells directly. γδ T cells express various activating receptors, such as NKG2D, which can bind to stress-induced ligands (e.g., MICA/B, ULBP) expressed on tumor cells. This interaction triggers the release of perforin and granzymes, which mediate the cytotoxicity of γδ T cells against GBM cells. Additionally, γδ T cells produce pro-inflammatory cytokines such as IFN-γ and TNF-α, which enhance their anti-tumor activity by promoting the activation of other immune cells, including CD8^+^ T cells and NK cells, involved in the anti-tumor immune response [12]. Moreover, γδ T cells can infiltrate GBM tumors and persist in the microenvironment, where they continue to exert their cytotoxic effects (Figure 1). In preclinical models, the adoptive transfer of γδ T cells has been shown to enhance survival and reduce tumor burden in GBM-bearing mice, further supporting their potential as an effective therapeutic option [33]. Importantly, γδ T cells can target tumor cells independently of MHC restriction, giving them an advantage over other immune cells that rely on antigen presentation for activation. This feature is essential in the context of GBM, where tumor cells often downregulate MHC molecules to evade recognition by conventional T cells [34]. Additionally, combining γδ T cell-based therapies with standard treatments such as TMZ has demonstrated synergistic effects, enhancing the therapeutic response and overcoming resistance mechanisms commonly observed in GBM treatment [30]. Recent studies have focused on using γδ T cells in combination with ICIs, such as PD-1/PD-L1 blockers, to further enhance their antitumor effects. These studies have shown that the blockade of immune checkpoint pathways can reactivate γδ T cells within the TME, promoting their anti-tumor activity and improving overall survival rates in animal models of GBM. Furthermore, γδ T cells exhibit a unique capacity for tissue infiltration, including the ability to cross the BBB, which makes them an attractive candidate for targeting CNS tumors like GBM. Taken together, these findings highlight the potential of γδ T cells as both a direct and indirect immune effector against GBM, and underscore the importance of advancing γδ T cell-based immunotherapies for GBM patients [27].

## 5. Pro-Tumoral Role of γδ T Cells in GBM

While γδ T cells are generally acknowledged for their role in mediating antitumor immunity, recent studies have shown that they can contribute to tumor progression in specific contexts, particularly in the immunosuppressive microenvironment of GBM. The GBM microenvironment is highly hostile to immune cells, characterized by the presence of various cytokines, including IL-1β, IL-6, TGF-β, and IL-23 [35]. These factors can significantly alter the functional behavior of γδ T cells, driving them to adopt a pro-tumoral, type-3 (Th17-like) phenotype (Figure 1). Although Vγ9Vδ2 T cells are known for their cytotoxic function, they display a dual role. High concentrations of TGF-β divert cells toward pro-tumor subtypes due to the downregulation of NKG2D and the expression of genes encoding granzymes and perforins [16]. Under such conditions, γδ T cells can differentiate into IL-17-producing cells [36] that support tumor progression; the production of IL-17 by these cells has been associated with the recruitment of immunosuppressive cells, including MDSCs and Tregs, which can dampen the efficacy of other immune responses within the tumor [28]. Furthermore, IL-17 can induce angiogenesis by stimulating endothelial cell proliferation and the secretion of pro-angiogenic factors, thereby facilitating the formation of new blood vessels that support tumor growth and metastasis. Another critical aspect of the pro-tumoral role of γδ T cells in GBM is their potential to differentiate into γδ Tregs, a subset of regulatory T cells that contribute to the suppression of immune responses in the TME. γδ Tregs exert their immunosuppressive effects by inhibiting dendritic cell maturation and impairing the activation of cytotoxic T cells and NK cells. This suppressive action further weakens the overall anti-tumor immune response and gives the tumor an advantage in evading immune surveillance. Furthermore, γδ T cells may be recruited to GBM lesions, where they secrete various cytokines that suppress the function of other immune cell populations, potentially aiding in tumor evasion [16]. This dual nature of γδ T cells highlights the complexity of their involvement in GBM and emphasizes the need for further research to clarify their precise role in tumor progression.

## 6. γδ T Cells in Cancer Immunotherapy

In recent years, immunotherapy has revolutionized cancer treatment by enhancing the immune system’s ability to eliminate malignant cells. Most immunotherapeutic approaches rely on αβ T lymphocytes, which, although highly specific, are limited by their dependence on MHC restriction, co-stimulatory signals, and recognition of tumor-associated antigens (TAAs) [37]. Many tumors, however, can escape αβ T cell-mediated immune surveillance by downregulating MHC class I molecules or losing antigen expression, thereby limiting the effectiveness of αβ T cell-based therapies [38,39]. γδ T cells provide a promising alternative due to their unique biology. Unlike αβ T cells, γδ T cells recognize stress-induced or transformation-associated antigens in an MHC-independent manner and do not require classical co-stimulation for activation [40]. This allows them to identify and destroy tumor cells that would otherwise evade conventional immune responses. Moreover, γδ T cells exhibit a rapid and potent effector response, including direct cytotoxicity through perforin and granzyme release, production of pro-inflammatory cytokines such as IFN-γ and TNF-α, and modulation of the tumor microenvironment through interactions with dendritic cells, NK cells, and other immune effectors [41,42].

The most well-characterized γδ T cell subset in the context of cancer is the Vγ9Vδ2 population, which predominates in peripheral blood. These cells can be activated using PAgs—either naturally derived (e.g., isopentenyl-pyrophosphate (IPP)) or synthetic analogs (e.g., BrHPP)—or pharmacologically through nitrogen-containing bisphosphonates (N-BPs), such as zoledronic acid (ZOL) [43]. N-BPs inhibit farnesyl pyrophosphate synthase in the mevalonate pathway, leading to intracellular PAgs accumulation and potent activation of Vγ9Vδ2 T cells [44]. Two main approaches have emerged in the past: in vivo activation using N-BPs and IL-2, and adoptive transfer of ex vivo expanded Vγ9Vδ2 T cells. While in vivo stimulation is less complex from a clinical standpoint, its efficacy may be hindered by transient responses or induction of functional exhaustion following repeated stimulation [45]. In contrast, ex vivo expansion allows more precise control over activation, phenotype, and dosing of the infused γδ T cells. Thanks to their lack of genetic restriction and graft-versus-host effects, allogeneic γδ T cells offer the potential benefit of being a readily available treatment option, compared to autologous therapies, since there is no need for time-consuming cell isolation and expansion procedures from the patient’s own cells. (Figure 2) [46]. Additionally, γδ T cells are genetically engineered to express chimeric antigen receptors (CARs), combining their natural cytotoxic properties with antigen-specific targeting (Figure 3). These CAR-γδ T cells have shown promise in preclinical studies for both hematologic and solid tumors [47,48]. Although Vγ9Vδ2 T cells are widely used in clinical settings due to their abundance and expandability, Vδ1^+^ γδ T cells are gaining attention. These cells are enriched in epithelial and mucosal tissues, exhibit a strong cytotoxic capacity, and are more resistant to activation-induced cell death, making them ideal candidates for durable antitumor responses [49]. However, the lack of standardized protocols for their isolation and expansion has historically limited their clinical application [50]. Recent developments have enabled the efficient expansion of Vδ1^+^ cells with potent cytotoxic profiles, making them suitable for adoptive therapy [51].

Furthermore, novel strategies are being explored to engage γδ T cells more selectively and effectively (Figure 2). These include bispecific γδ T cell engagers that simultaneously bind to tumor antigens and γδ TCR chains, thereby redirecting γδ T cell cytotoxicity toward tumor cells [52]. For instance, clinical trials investigating bispecific antibodies such as LAVA-051 and LAVA-1207 are currently underway in hematologic and solid tumors (NCT05369000). Adoptive transfer of ex vivo expanded γδ T cells has shown promising efficacy in preclinical models of GBM, melanoma, and other malignancies, with robust tumor infiltration and cytotoxicity (NCT04165941). However, challenges remain, including variability in patient responses and the need for optimal dosing strategies. Other companies are developing allogeneic CAR-γδ T cells, leveraging their lower risk of graft-versus-host disease and reduced incidence of cytokine release syndrome compared to αβ CAR-T cells [48]. While αβ T cell-based therapies, such as CAR-αβ T cells, have achieved remarkable success in hematologic malignancies, their efficacy in solid tumors is limited by antigen heterogeneity and the immunosuppressive TME [39,47]. ICIs targeting PD-1/PD-L1 or CTLA-4 have revolutionized treatment for certain cancers but rely on pre-existing tumor-specific T cell responses, which may be absent in immunologically “cold” tumors [37]. In contrast, γδ T cells’ ability to target stress-induced antigens without MHC restriction makes them an attractive option for overcoming these limitations [40]. Furthermore, the genetic engineering of CAR-γδ T cells combines innate and adaptive immune advantages, potentially addressing antigen escape mechanisms more effectively than conventional CAR-T cells [41].

Despite their promise, γδ T cell-based therapies face several hurdles:•Exhaustion: Repeated stimulation or chronic activation can lead to functional exhaustion, which reduces efficacy over time [53].•Off-Target Effects: The MHC-independent recognition of antigens raises concerns about potential off-tumor cytotoxicity [54].•Tumor Heterogeneity: Variability in the expression of stress-induced antigens across tumors may limit the applicability of γδ T cell therapies [49].•Expansion Variability: Ex vivo expansion protocols often yield inconsistent results, which can impact the scalability and reproducibility of therapies [50].

Addressing these challenges requires further optimization of expansion protocols, improved understanding of γδ T cell biology, and the development of strategies to mitigate exhaustion and off-target effects. In fact, several biotech companies are at the forefront of advancing γδ T cell therapies. Lava Therapeutics (Utrecht, The Netherlands) is developing bispecific γδ T cell engagers, with LAVA-051 and LAVA-1207 in clinical trials (NCT05369000). Adicet Bio (Boston, MA, USA) is exploring allogeneic CAR-γδ T cells to minimize graft-versus-host disease (NCT04735471). IN8bio (New York, NY, USA) focuses on utilizing γδ T cells for the treatment of solid tumors, with promising early-phase trial results (NCT04165941). GammaDelta Therapeutics (London, UK) is pioneering methods to enhance the expansion and efficacy of Vδ1^+^ γδ T cells [51]. These companies’ innovations highlight the growing interest and investment in γδ T cell-based therapies. Several other biotech firms are making significant strides in advancing γδ T cell therapies. TC BioPharm (Motherwell, UK) is conducting two investigator-initiated clinical trials for its unmodified γδ T cell product line, including a Phase 2b/3 pivotal trial in the treatment of acute myeloid leukemia using the company’s proprietary allogeneic CryoTC technology (Edinburgh, UK) (NCT04735471). CytoMed Therapeutics (Singapore City, Singapore) has initiated the ANGELICA Trial (NCT05653271), a first-in-human Phase I clinical trial evaluating allogeneic NKG2DL-targeting CAR-γδ T cells (CTM-N2D) in patients with advanced solid tumors or hematological malignancies. Acepodia (Alameda, CA, USA) is developing ACE1831, an off-the-shelf γδ T cell therapy currently undergoing a Phase I clinical trial (NCT05653271) for the treatment of non-Hodgkin’s lymphoma. Additionally, Expression Therapeutics (Tucker, GA, USA) is developing a γδ T cell therapy that, when used in conjunction with standard chemotherapy and immunotherapy, aims to effectively eradicate neuroblastoma tumor cells. These efforts reflect the wide range of strategies aimed at fully leveraging the therapeutic potential of γδ T cells.

## 7. γδ T Cells in GBM Immunotherapy

GBM continues to pose a significant therapeutic challenge, with standard treatments such as surgery, radiotherapy, and chemotherapy offering limited survival benefits [55,56]. In this context, γδ T cells have emerged as a promising immunotherapeutic strategy due to their unique properties, including MHC-unrestricted target cell recognition and an immediate response to stress-induced ligands expressed by tumor cells [57,58]. Among the γδ T cell subsets, Vγ9Vδ2 T cells have demonstrated notable antitumor activity against GBM through mechanisms such as perforin/granzyme-mediated lysis and NKG2D-dependent recognition of ligands including ULBP and MIC [15,59,60]. These ligands are upregulated on tumor cells under stress conditions and are further enhanced by treatments including TMZ and ZOL. ZOL plays a dual role in GBM immunotherapy by inducing the accumulation of isopentenyl-pyrophosphate, a ligand for Vγ9Vδ2 T cells, while also sensitizing glioblastoma cells to immune attack and increasing the surface expression of NKG2D ligands, rendering glioblastoma cells more susceptible to γδ T cell-mediated cytotoxicity [60,61]. However, the toxic effects of TMZ on γδ T cells themselves necessitate innovative solutions such as genetic engineering to confer resistance. For instance, γδ T cells modified with lentiviral vectors encoding O6-methylguanine-DNA-methyltransferase (MGMT) retain full functionality in the presence of TMZ, allowing for their concurrent use with chemotherapeutic regimens [62]. Ex vivo expansion of γδ T cells from peripheral blood has proven reliable to generate large numbers of functional cells capable of targeting GBM. These expanded cells have demonstrated significant cytotoxicity against autologous tumor cells while sparing normal tissues, thereby showcasing their safety profile [56,63]. Preclinical models have further validated the efficacy of γδ T cells, with intracranial infusions slowing tumor progression and extending survival in xenograft models [32]. Moreover, combinations of cytokines such as IL-2, IL-12, and IL-15 have synergistically enhanced the proliferation and cytotoxic activity of γδ T cells, offering additional avenues for therapeutic optimization [55,57,58]. Despite their promise, γδ T cells—like other cellular therapies—must overcome the profoundly immunosuppressive TME of GBM, which is characterized by Tregs, MDSCs, and TAMs that secrete inhibitory cytokines such as IL-10 and TGF-β [64]. A notable example of the clinical translation of genetically modified γδ T cells is the INB-200 Phase 1 clinical trial, currently underway and sponsored by IN8bio. This study evaluates the safety, tolerability, and potential efficacy of intracranial infusion of autologous, ex vivo expanded γδ T cells in patients with newly diagnosed or recurrent GBM, in combination with TMZ. The cells are administered post-tumor resection alongside standard TMZ therapy, leveraging the drug’s tumor-sensitizing effects without impairing the function of the infused γδ T cells. Unlike conventional CAR-T approaches, this strategy exploits the innate antitumor properties of γδ T cells, which are enhanced through expansion and genetic manipulation, while maintaining a favorable safety profile. Preliminary data released by IN8bio indicate that the infusions are well tolerated, with no treatment-related serious adverse events and early signs of antitumor activity. This trial represents one of the first clinical efforts to validate γδ T cells as a viable form of cellular immunotherapy for GBM, laying the groundwork for future large-scale, controlled studies (NCT04165941, NCT03533816). Additionally, the integration of γδ T cells into combinatorial immunotherapy regimens—including checkpoint inhibitors, NK cell therapies, and dendritic cell vaccines—is under investigation, as synergistic approaches may help reverse TME-mediated immune suppression and improve treatment outcomes [64]. Despite these advances, the immunosuppressive TME of GBM remains a significant barrier. This microenvironment, characterized by inhibitory cytokines and a lack of effective lymphocyte infiltration, limits the full potential of γδ T cell-based therapies [65]. Innovative strategies are being explored to overcome these challenges, including the use of BTN3A1 agonistic monoclonal antibodies and developing CAR γδ T cells targeting GBM-associated antigens. CAR-engineered γδ T cells have shown remarkable preclinical efficacy, highlighting their ability to enhance tumor recognition and destruction [66].

Furthermore, studies have underscored the versatility of γδ T cells in adapting to the complex GBM microenvironment. For instance, γδ T cells co-cultured with GBM cells exhibit a Th1-like profile characterized by increased IFN-γ and TNF-α production, suggesting their ability to reshape the immune landscape within tumors [67]. The infiltration of unique γδ T cell subsets, such as Vγ9Jγ2-Vδ2 T cells, into GBM tissues further emphasizes their tumor-specific adaptations. These cells, distinct from their peripheral counterparts, demonstrate robust antitumor responses, offering insights into their potential for precision immunotherapy [68]. Moreover, the functional reprogramming of γδ T cells in the TME, resulting in a tumor-promoting phenotype, has been reported to occur for Vδ1 T cells, but not for the Vγ9Vδ2 subset. In fact, there is no evidence that Vγ9Vδ2 T cells that have been expanded ex vivo for subsequent in vivo use are susceptible to reprogramming to a pro-tumor phenotype. Therefore, these intrinsic properties of Vγ9Vδ2 T cells strongly limit the possible immunosuppressive activities of the TME and provide a way to sustain tumor control.

Overall, γδ T cells represent a powerful tool in the fight against GBM, combining intrinsic cytotoxicity with innovative engineering approaches to address the challenges posed by this formidable tumor. As research continues to refine these therapies, γδ T cells are poised to play a central role in developing next-generation immunotherapies for GBM, offering hope for improved outcomes in this devastating disease.

Table 1 integrates findings from experimental, preclinical, and early-phase clinical studies selected based on their relevance to the functional characterization and therapeutic exploitation of γδ T cells in GBM. Although this review was conducted using a narrative approach, we applied a structured literature search as illustrated in the adapted PRISMA-style diagram (Figure 4). To identify relevant literature, a comprehensive but non-systematic search was performed on PubMed/MEDLINE, Scopus, and Web of Science databases including a combination of the following keywords: “γδ T cells” OR “gamma delta T lymphocytes” AND “glioblastoma” OR “GBM” OR “glioblastoma multiforme”. Searches were limited to English-language, peer-reviewed articles, with no publication date restrictions. Priority was given to studies published between 2010 and 2024 due to the growing interest in the topic. Additional references were retrieved by manually screening the bibliographies of relevant articles. Specifically, we included original experimental (in vitro/in vivo) studies investigating γδ T cell activity in GBM, clinical studies assessing the safety or efficacy of γδ T cell-based therapies in glioblastoma patients, and reports on γδ T cell engineering strategies (e.g., CAR-γδ T cells) applicable to GBM. Conversely, we excluded studies involving γδ T cells in cancers other than GBM without mechanistic overlap, reports not involving human or murine GBM models, non-English or non-peer-reviewed literature (e.g., abstracts, editorials), and reviews or conceptual papers providing mechanistic insights relevant to GBM.

Data were qualitatively extracted and organized thematically, and the studies were categorized based on:-γδ T cell subtypes and functional roles (pro-tumoral vs. anti-tumoral),-Model system (in vitro, murine, human),-Therapeutic approach (e.g., zoledronic acid, checkpoint inhibitors, CAR-T constructs),-Route of administration and outcome measures (tumor response, survival, immunological profiling).

No quantitative meta-analysis or formal quality assessment was conducted, in line with the narrative nature of this review.

## 8. Conclusions

GBM is a highly aggressive and complex brain tumor, characterized by an immunosuppressive microenvironment that limits the efficacy of conventional therapies and contributes to its highly unfavorable prognosis. Immunotherapy represents a promising approach to overcome these limitations, and γδ T cells, with their ability to recognize tumor cells independently of MHC, are emerging as an interesting strategy. However, the application of immunotherapy in GBM presents unique challenges, related to the complexity of the TME and the need to overcome the blood–brain barrier. γδ T cells offer potential to address these challenges, owing to their intrinsic cytotoxicity and the capacity to modulate the immune response. Research focuses on developing strategies to enhance their antitumor activity and the capacity to infiltrate the tumor. Despite promising preclinical results and early-phase clinical trials, such as the INB-200 Phase 1 study of gene-modified autologous γδ T cells [75] and intraventricular administration of CARv3-TEAM-E T cells [76], the clinical translation of γδ T cell-based immunotherapy for GBM is still in its early stages. Further studies are necessary to optimize therapeutic strategies, fully understand the role of these cells in the context of GBM, and develop novel technologies, such as CAR-γδ T cells and bispecific constructs, to improve their efficacy and specificity [74,76]. In conclusion, γδ T cell immunotherapy offers a concrete prospect for treating GBM, but its full potential requires further investigation and innovation to overcome the challenges posed by this devastating tumor.

## Figures and Tables

**Figure 1 cancers-17-02660-f001:**
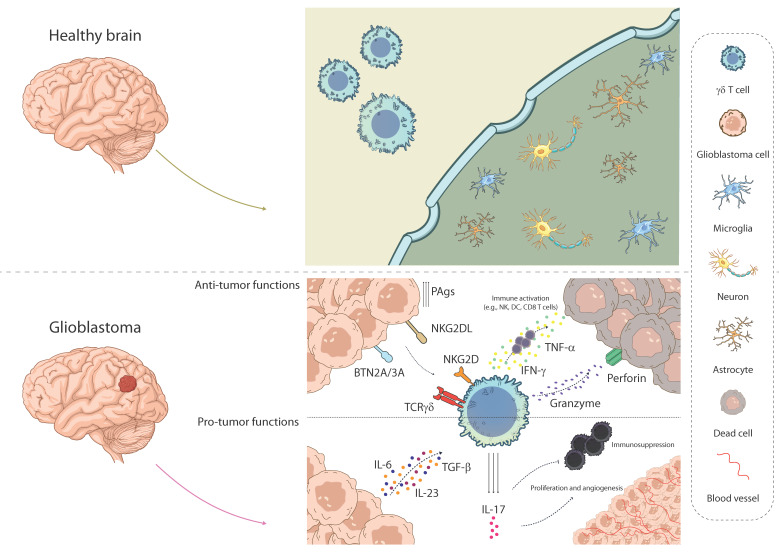
γδ T cell responses in the healthy brain and GBM microenvironment. Comparison of the immune landscape in the healthy CNS (**top**) and GBM (**bottom**). In homeostasis, γδ T cells are largely excluded by the blood–brain barrier, and the parenchyma contains mainly microglia, astrocytes, and neurons. In GBM, γδ T cells infiltrate the tumor, where they can exert anti-tumor functions via TCRγδ, NKG2D, and BTN2A/3A recognition, releasing IFN-γ, TNF-α, perforin, and granzymes. However, GBM-derived cytokines such as IL-6, TGF-β, and IL-23 can shift γδ T cells toward a pro-tumoral IL-17–producing phenotype that promotes angiogenesis. Illustration from NIAID NIH BIOART Source.

**Figure 2 cancers-17-02660-f002:**
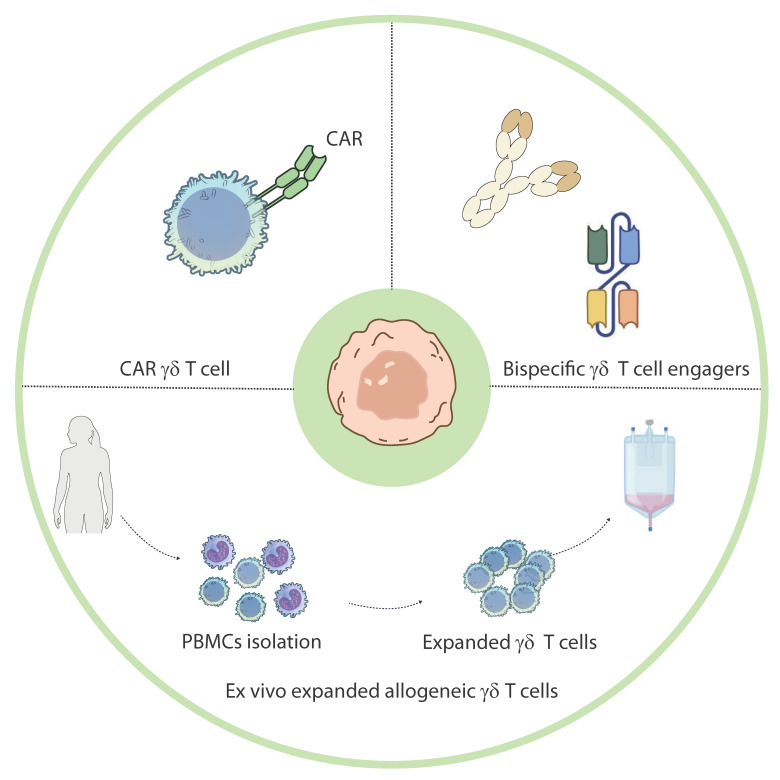
γδ T cell-based immunotherapeutic strategies for GBM. Overview of current γδ T cell-based therapeutic approaches for GBM. These include adoptive transfer of ex vivo expanded allogeneic γδ T cells, bispecific engagers that redirect γδ T cells toward tumor cells, and CAR-engineered γδ T cells designed to enhance specificity and cytotoxicity. These strategies aim to harness the innate anti-tumor functions of γδ T cells while overcoming the immunosuppressive glioblastoma microenvironment. Illustration from NIAID NIH BIOART Source.

**Figure 3 cancers-17-02660-f003:**
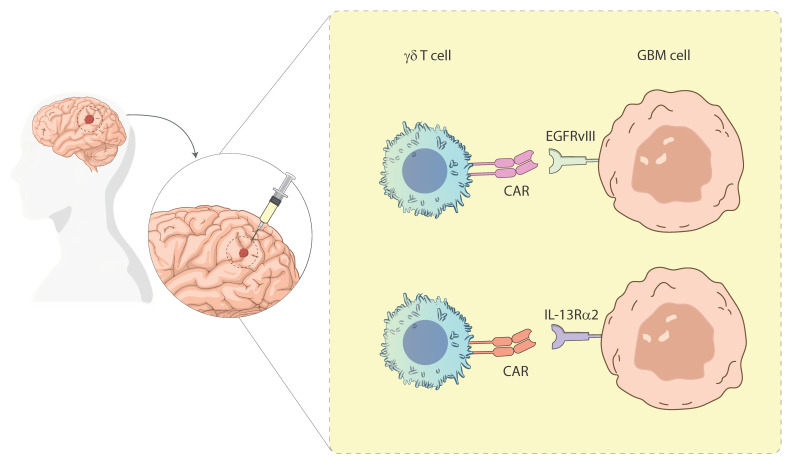
Targeted CAR γδ T cell therapies for GBM. Schematic representation of γδ T cells engineered with CARs targeting GBM. The image illustrates γδ T cells expressing CARs specific for the tumor-associated antigens EGFRvIII (Epidermal Growth Factor) and IL-13Rα2, engaging with GBM cells. This strategy highlights the potential of γδ T cells as an alternative platform for CAR-based immunotherapy in the treatment of GBM. Illustration from NIAID NIH BIOART Source.

**Figure 4 cancers-17-02660-f004:**
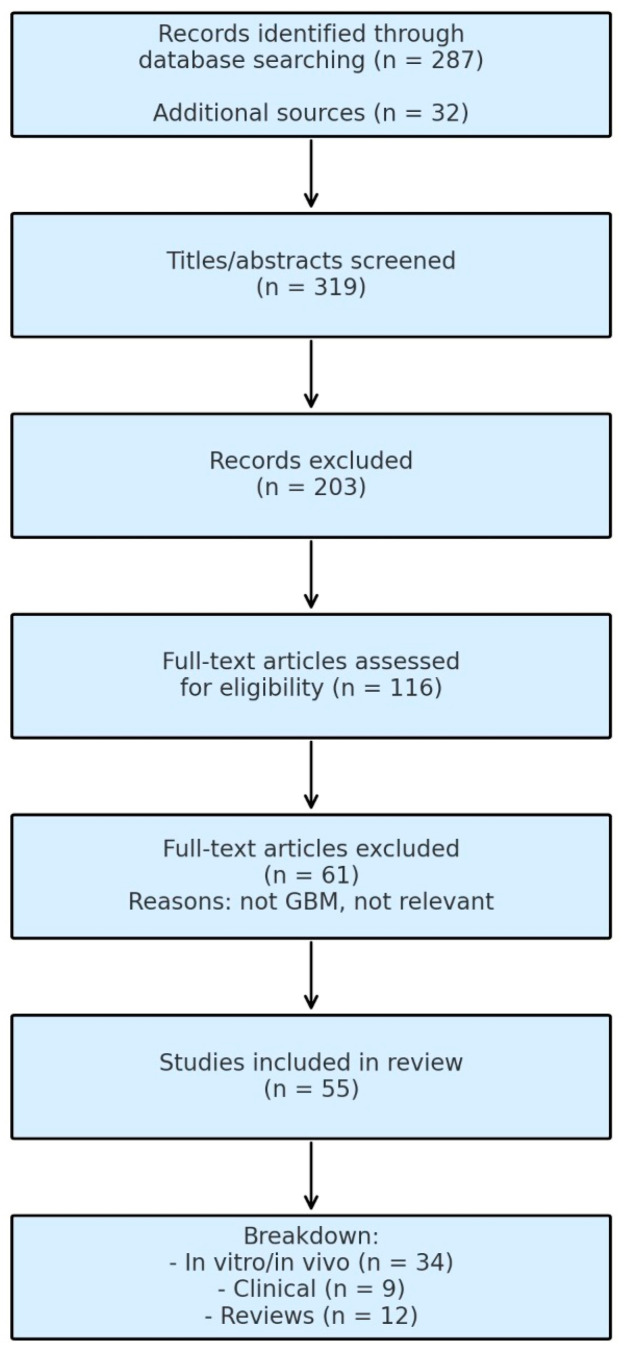
Summary of study selection process. This diagram is illustrative and reflects the authors’ narrative selection process rather than a formal PRISMA workflow.

**Table 1 cancers-17-02660-t001:** Summary of key studies investigating the therapeutic use of γδ T cells in GBM. The table highlights study characteristics, including type, aim, techniques, main findings, administration route, target, adverse events, and patient population. [Abbreviations: ROA-Administration Route; AE-Adverse Events; N/A-Not Available; IC-Intracranial; ST-IC-Stereotactic Intracranial; IP-Intraperitoneal; IT-Intra-tumoral; ICV-Intraventricular].

Author, Year	Study Type	Aim of the Study	Technique	Findings	ROA (if Therapy)	Target	AE	Patients
Bryant et al., 2009 [69]	In vitro (preclinical)	Assess the innate immune function of T cells in GBM and their therapeutic potency	T cells from GBM patients and healthy controls were expanded and tested against U251MG, D54MG, U373MG, U87MG, and primary GBM explants	GBM patients had fewer and weaker T cells. Activated T cells could kill GBM cells but spared normal astrocytes, indicating selective cytotoxicity.	N/A	GBM cell lines and primary cultures	N/A	GBM patients and healthy controls
Bryant et al., 2011 [15]	In vivo (mouse xenograft)	Evaluate T cell migration, infiltration, and antitumor efficacy in GBM	U251MG cells were implanted intracranially in immunodeficient mice, followed by stereotactic injection of expanded T cells	T cells significantly slowed tumor progression and extended survival. Demonstrated in vivo trafficking and tumor infiltration.	IC	U251MG xenografts	N/A	Immunodeficient mouse model
Cimini et al., 2011 [60]	In vitro	Investigate whether Zol enhances γδ T cell killing of GBM	Vδ2 T cells activated with phosphoantigens were co-cultured with T70, U251, and U373 GBM cells ± Zol	Zol enhanced perforin-mediated killing and induced apoptosis in a dose-dependent manner, increasing γδ T cell efficacy.	N/A	GBM cell lines	N/A	Cell line model
Lamb et al., 2013 [62]	In vitro	Determine if MGMT-modified T cells resist TMZ and retain cytotoxicity	T cells were transduced with MGMT and tested on TMZ-resistant U87, U373, SNB-19 GBM lines	MGMT+ T cells resisted TMZ and remained effective against GBM. TMZ increased NKG2D ligands, enhancing T cell sensitivity.	N/A	TMZ-resistant GBM cells	N/A	GBM cell lines
Nakazawa et al., 2014 [61]	In vitro	Examine the role of Zol in boosting γδ T cell cytotoxicity	U87MG, U138MG, A172 GBM lines were treated with Zol before γδ T cell exposure	Zol sensitized GBM cells, killing from <32% to over 80% in some lines. Blocked by anti-TCR antibody.	N/A	GBM cell lines pretreated with Zol	N/A	GBM cell lines
Beck et al., 2015 [70]	In vivo (immunocompetent mouse)	Characterize γδ T cell kinetics and immune suppression in glioma	GL261 murine glioma model used to monitor T cell cytokines and tumor progression	T cells expanded early, then collapsed. Tumors secreted TGF-β, promoting immune evasion. No survival difference in γδ T cell KO mice.	N/A	GL261 gliomas	N/A	Immunocompetent mice
Chitadze et al., 2016 [71]	In vitro	Assess impact of TMZ and metalloprotease inhibitors on NKG2DL expression	U87MG, A172, T98G, U251MG treated with TMZ and ADAM10/17 inhibitors; tested with BrHPP-activated γδ T cells	TMZ upregulated ULBP2 on GBM surface. ADAM inhibition prevented shedding, enhancing T cell-mediated lysis.	N/A	NKG2DLs (MICA, MICB, ULBPs)	N/A	Human GBM cell lines
Jarry et al., 2016 [72]	In vivo (xenograft)	Test if stereotactic injection of γδ T cells with Zol improves survival	Vγ9Vδ2 T cells were injected into brains of mice bearing U87MG or GBM-10 xenografts ± Zol	Stereotactic delivery led to tumor clearance and improved survival, especially with Zol sensitization.	ST-IC	U87MG, GBM-10	N/A	Mice with human GBM xenografts
Nakazawa et al., 2016 [73]	In vitro & in vivo	Evaluate synergy between minodronate and γδ T cells	Co-culture of GBM cells with γδ T cells ± minodronate; IP injection in immunocompromised mice	Combination induced higher granzyme B, TNF-α release, stronger apoptosis, and tumor inhibition in vivo.	IP (mice)	U87MG, U138MG	N/A	Immunodeficient mouse model
Joalland et al., 2018 [64]	In vivo	Investigate effect of IL-21 on γδ T cell cytotoxicity in GBM	Vγ9Vδ2 T cells were pretreated with IL-21 and injected into GBM-bearing mice	IL-21 enhanced cytotoxicity against GBM-1 and U87MG. Survival extended from 41 to 66 days.	ST-IC	GBM-1 (primary), U87MG	N/A	Orthotopic GBM mouse model
Chauvin et al., 2019 [74]	In vitro & in vivo	Evaluate immunoreactivity of Vγ9Vδ2 T cells against molecular subtypes of patient-derived GBM cells	Co-culture of Vγ9Vδ2 T cells with mesenchymal (GBM-1) and classical/proneural (GBM-10) cells; tested with/without nitrogen bisphosphonate	Vγ9Vδ2 T cells naturally lysed mesenchymal GBM-1 but not GBM-10. N-BPs were required to sensitize GBM-10. Killing depended on TCR and NKG2D pathways.	N/A	Mesenchymal vs. classical GBM subtypes	N/A	Patient-derived GBM primary cultures
Lee et al., 2019 [33]	In vitro	Characterize tumor-infiltrating γδ T cells in GBM patients	TCR sequencing of tumor and blood-derived γδ T cells from patients	Tumor-infiltrating Vγ9Vδ2 T cells had non-canonical TCR sequences and cytotoxic gene profiles similar to Th1 and M1 macrophages.	N/A	Tumor-infiltrating Vγ9Vδ2 T cells	N/A	GBM patients
Choi et al., 2023 [75]	Review	Summarize γδ T cells as a non-MHC-based GBM therapy	Review of existing in vitro/in vivo and early clinical data	Vγ9Vδ2 T cells use NKG2D/DNAM-1 to recognize stress ligands; CAR or checkpoint-enhanced platforms proposed.	IT, ICV (proposed)	NKG2D, DNAM-1 ligands	N/A	Preclinical evidence
Nabors et al., 2024 [63]	Phase 1 clinical trial (ongoing)	Test safety/feasibility of MGMT-modified γδ T cells with TMZ	Drug-resistant immunotherapy: MGMT+ γδ T cells injected during concurrent TMZ	MGMT expression allows T cells to persist during TMZ. Trial ongoing; preliminary data shows feasibility.	IC	NKG2D ligands (e.g., MICA, ULBP2)	Not fully disclosed	Newly diagnosed GBM patients
Lobbous et al., 2024 [76]	Phase 1 trial	Determine safety of intracranial MGMT+ γδ T cells during SOC	MGMT+ γδ T cells injected into surgical resection cavity during standard Stupp regimen	No dose-limiting toxicities. Some patients had PFS benefit. Mild hematological AEs only.	IC (resection cavity)	NKG2D ligands	Mild hematologic; no CRS	Newly diagnosed GBM patients
Choi et al., 2024 [77]	Phase 1 trial (dose escalation)	Assess safety of CARv3-TEAM-E T cells targeting EGFR/EGFRvIII	Autologous CAR-T cells delivered intraventricularly; dual-targeting with enhanced cytotoxic modules	Partial tumor regressions observed. No dose-limiting toxicities. Reversible neurotoxicity (grade 3) and fatigue reported.	ICV	EGFRvIII and wild-type EGFR	Grade 3 encephalopathy, fatigue	Recurrent EGFRvIII+ GBM patients

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
