# Peer review of "γδ T Cells in Glioblastoma Multiforme: Novel Roles and Therapeutic Opportunities"

_cancers, 2025, doi:10.3390/cancers17162660_

Round 1
Reviewer 1 Report
Comments and Suggestions for Authors
This is a well-organized review that highlights both the opportunities and challenges associated with γδ T cell-based strategies, especially for GBM.
- 1. since T cells are unfamiliar to many readers, it should be noted how much T cells normally infiltrate GBM tissue.
- In the sections Human γδ T Cells in GBM, Anti-tumoral Role of γδ T Cells in GBM, and Pro-tumoral Role of γδ T Cells in GBM, a brief schema should be included for the reader's understanding.
- References 75 and 76 should also be cited with a brief explanation in the text.
- In introduction section, the usual abbreviation for temozolomide is TMZ, not TZM,
- Abstract section should be more detailed than Simple summary section.
Reviewer 2 Report
Comments and Suggestions for Authors
Thank you for your comments and review submitted to cancers.
Overall the topic of gamma/delta T-cells in GBM is an interesting topic. THe authors should make some chnages to make the review to present the topic from a more knowledgable perspective that provides a more in depth knowledge perspective than is currently being discussed.
For instance the authors give equal weight to discussing both pro and anti tumoral properties of g/d tcells in GBM.(pages 4 and 5). Given this presentation, it is unclear as to why clinical trials are being implemented using this technology, unless the majority consensus indicates g/d t cells are indeed function more as tumor suppressors.
This seems to be the case, as the table in the manuscript clearly indicates many anti-tumorigenic properties of gd t cells in pre-clinical models
I think the authors should expand on the the rationale for why pro-tumorigenic events would occur with gd t-cells. perhaps due to a heightened and prolonged inflammatory response that results in tissue damage, inflammation, and a pro tumorigenic environment. And if this is the reason, then what are the mechanisms to prevent engagement of such pro tumorigenic events.
From a therapeutic perspective, it might be worth while to make a figure describing the chemistries of the therapy, such that one could understand how these new drugs are operating within the intracranial space.
From a mechanistic perspective, in might be worth while to make a figure describing the intracellular pathways involved when g/d t cells interact with a GBM cells versus a normal astrocyte.
Comments on the Quality of English LanguageThank you for your comments and review submitted to cancers.
Overall the topic of gamma/delta T-cells in GBM is an interesting topic. THe authors should make some chnages to make the review to present the topic from a more knowledgable perspective that provides a more in depth knowledge perspective than is currently being discussed.
For instance the authors give equal weight to discussing both pro and anti tumoral properties of g/d tcells in GBM.(pages 4 and 5). Given this presentation, it is unclear as to why clinical trials are being implemented using this technology, unless the majority consensus indicates g/d t cells are indeed function more as tumor suppressors.
This seems to be the case, as the table in the manuscript clearly indicates many anti-tumorigenic properties of gd t cells in pre-clinical models
I think the authors should expand on the the rationale for why pro-tumorigenic events would occur with gd t-cells. perhaps due to a heightened and prolonged inflammatory response that results in tissue damage, inflammation, and a pro tumorigenic environment. And if this is the reason, then what are the mechanisms to prevent engagement of such pro tumorigenic events.
From a therapeutic perspective, it might be worth while to make a figure describing the chemistries of the therapy, such that one could understand how these new drugs are operating within the intracranial space.
From a mechanistic perspective, in might be worth while to make a figure describing the intracellular pathways involved when g/d t cells interact with a GBM cells versus a normal astrocyte.
Reviewer 3 Report
Comments and Suggestions for Authors
The review article is interesting and scientifically valuable. However, it can benefit from better organizational structure. My suggestions are:
- Introduction section is too long and contains a lot of references. To make it easy to understand please divide it into subsections and discuss the relevant literature in respective area. Especially immune checkpoint inhibitors (ICIs) related write up can be presented separately. Similarly, γδ T cells interaction with other immune cells can also be presented separately.
- An illustration for "γδ T cell characteristics and functions" will be helpful for readers to understand this concept.
- The contradictory role of γδ T cells with respect to their anti-tumoral as well as pro-tumoral activity is unclear. GBM is has poor prognosis due to its immunosuppressive TME. Does γδ T cells provide any advantage over this ?
- The key studies included in the table need to be defined. What criteria was used to select the presented studies. Did you include all studies related to the topic or just a subset of it and why.
Round 2
Reviewer 2 Report
Comments and Suggestions for Authors.